# Optical and Thermal Design and Analysis of Phase-Change Metalenses for Active Numerical Aperture Control

**DOI:** 10.3390/nano12152689

**Published:** 2022-08-05

**Authors:** George Braid, Carlota Ruiz de Galarreta, Andrew Comley, Jacopo Bertolotti, C. David Wright

**Affiliations:** 1College of Engineering Mathematics and Physical Sciences, University of Exeter, Exeter EX4 4QF, UK; 2AWE Aldermaston, Reading RG7 4PR, UK

**Keywords:** active lenses, active metasurfaces, phase-change metasurfaces

## Abstract

The control of a lens’s numerical aperture has potential applications in areas such as photography and imaging, displays, sensing, laser processing and even laser-implosion fusion. In such fields, the ability to control lens properties dynamically is of much interest, and active meta-lenses of various kinds are under investigation due to their modulation speed and compactness. However, as of yet, meta-lenses that explicitly offer dynamic control of a lens’s numerical aperture have received little attention. Here, we design and simulate active meta-lenses (specifically, focusing meta-mirrors) using chalcogenide phase-change materials to provide such control. We show that, operating at a wavelength of 3000 nm, our devices can change the numerical aperture by up to a factor of 1.85 and operate at optical intensities of the order of 1.2 × 10^9^ Wm^−2^. Furthermore, we show the scalability of our design towards shorter wavelengths (visible spectrum), where we demonstrate a change in NA by a factor of 1.92.

## 1. Introduction

Optical metasurfaces are thin materials comprising arrays of sub-wavelength resonant structures (meta-atoms) whose geometry can be optimised to modulate the phase, amplitude and/or polarisation of incident light [1,2]. This allows the output wavefront to be tailored to a particular requirement, and the small size facilitates easier integration into other systems than conventional optical components allow [1,3]. In recent years, these properties have prompted the development of various metasurface devices for global or local amplitude, phase and polarisation control, including perfect absorbers [4], beam deflectors [5], holograms [6], polarisers [7] and flat lenses [8]. However, the responses of such ‘conventional’ metasurfaces are static: their effect on incoming beams is fixed. Active metasurfaces, on the other hand, allow for dynamic manipulation of the metasurface’s optical response, greatly increasing the range of possible applications.

Examples of the functionality achieved by active metasurfaces include modulation of the optical absorptance, reflectance and transmittance [9,10]; beam steering [11]; dynamic focusing [12]; active polarisation control [13] and dynamic holography [14]. The amplitude modulation approaches have applications as optical switches for communications networks, integrated photonic circuits [15] or refractive index sensors [16], as well as in thermal emission control [17]. Active lensing can entail moving the focal spot both along the axis and laterally. The former is useful for scanning depth in biomedical imaging [18] and wavefront sensing [19]. Applications for the latter include directed beams from telecommunications antennas [20], selecting the destinations of optical interconnects in data centre networks [21] and beam scanning in LIDAR [22]. Motivated by such applications, many different approaches to the dynamic control of metasurfaces have been investigated. Some of these are based on mechanical movement of components [23,24], but the vast majority are based on combining metasurfaces with materials whose refractive index can be selectively tailored via different mechanisms. These include the use of liquid crystals [25], materials with high thermo-optical dispersion [26], chemical reactions such as hydrogenation or de-hydrogenation [27], optical non-linearities such as the Kerr effect [28], modifications to charge carrier density through optical [29] or electrical signals [30] or, the approach used in this paper, phase-change materials (PCMs), which include VO${}_{2}$ [31,32] and chalcogenide PCMs [12,33,34,35,36,37,38]. We use chalcogenide PCMs, which we choose over VO${}_{2}$ for their non-volatility, since we consider that non-volatility offers benefits for this particular application. PCMs have two solid phases (amorphous and crystalline) with substantially different physical properties, including the refractive index. Each of these phases is stable at room temperature (i.e., non-volatile) but can be converted to the other by a thermal impulse (supplied electrically or optically) [39]. This conversion can be completed on nanosecond timescales, making PCMs attractive for dynamic metasurface designs having high speed and low power consumption [40]. Intermediate states, lying between the fully amorphous and fully crystalline, can also be achieved, leading to additional functionality.

Much work has already been done investigating PCM metasurfaces offering dynamic control of various output beam properties [9,10,11,12,13,14,33,34,35,36,37,38,41,42]. However, to date, the active control of a lens’s numerical aperture employing phase-change metasurfaces has not been reported, to the best of our knowledge. Numerical aperture (NA) control is of key importance in photography and imaging, a field in which there is much recent interest in the application of metasurfaces [43,44,45,46]. It may also be useful for laser processing of sophisticated materials such as biomimetic surfaces and metasurfaces [47,48] or for chemical sensing [49]. Numerical aperture control might also find applications in the challenging field of inertial confinement fusion, where lasers are used to compress millimetre-sized capsules filled with deuterium–tritium fuel: as the fuel pellet implodes, it is important to maintain efficient transfer of energy from the laser to the pellet, and this could be facilitated by the control of a focused spot size (NA) [50].

In this paper, therefore, we present the design of phase-change-material-based meta-lenses (specifically focusing meta-reflectors) with dynamic NA control, exploring two different spectral regimes (visible and mid-wave infrared). In view of the anticipated applications, we concentrate our efforts on longer focal lengths (hence, in some cases, small NAs), but the designs are adaptable to other cases by simply increasing the solid angle defined by the lens (i.e., via increasing the ratio between lens diameter and focal length, Ø/f). We do not rely on the Pancharatnam–Berry phase, but use polarisation-independent resonant elements. These resonators have small PCM volumes, facilitating the reversible phase transition [33,42], and they are a simple shape with a low aspect ratio, making fabrication easier. The performance of our devices in terms of efficiency and spot sizes has been analysed using diffraction theory, revealing achievable NA control by factors of around 1.8 to 1.9. Finally, since PCMs might, due to inadvertent heating, undergo unwanted changes in their structural phase when controlling high-power laser beams, the maximum incident laser intensities that our devices can handle under constant illumination have been also evaluated in order to assess their suitability for high power applications.

## 2. Materials and Methods

### 2.1. Working Principle of Active NA Control

Our metasurface design builds on arrays of hybrid dielectric/plasmonic resonators in which meta-atoms consist of sub-wavelength cylinders of a suitable dielectric having a thin PCM embedded within it, all placed on a metal back plane [33,42]. As generically depicted in Figure 1a, our metasurfaces are capable of tuning the NA for a fixed focal length (and therefore capable of actively tuning both the depth of focus and spot size). This is possible via selectively controlling the solid angle reaching the focal plane, which can be achieved by dividing our arrays in two concentric circular regions: an outer ring-like region, and an inner circular region. The inner region is always designed to bring light to a focus, which is achieved by arranging the local optical phase φ of the meta-atoms according to the following (parabolic) profile:(1)$$\varphi \left(x,y,\lambda \right)=\varphi \left(x=0,y=0,\lambda \right)-\frac{2\pi }{\lambda }\left(\sqrt{x^{2}+y^{2}+f^{2}}-f\right),$$
where *x* and *y* are spatial coordinates with the origin at the optical axis, *f* is the desired focal length and λ is the wavelength [51]. On the other hand, the outer region is designed to provide a spatially invariant (flat) phase profile (φ=const). To illustrate this concept, in Figure 1b we show an example of two different spatial phase profiles, yielding a high NA (left, no outer ring region) and low NA (right, outer ring region with nearly flat phase profile). To achieve this, resonators in one PCM state have been designed to provide a full 2π (360${}^{\circ }$) phase coverage to incident light via locally changing their diameters around the resonant frequency. When resonators are switched to the other PCM state, the resonant frequency is abruptly shifted due to a dramatic change in the PCM’s refractive index, resulting, for a properly designed array, in a flat optical phase profile. As depicted in Figure 1c, NA control can be therefore triggered by arranging amorphous and crystalline resonators or meta-atoms concentrically. The dielectric resonators are arranged in a square lattice lying on a metallic surface that acts as a reflective symmetry plane. The use of a metallic substrate brings several benefits in terms of practicability. From an optics point of view, working in reflection allows us to improve local phase accumulation around the resonance employing low aspect ratio resonators (which are easier to fabricate). In terms of thermal performance, having a back reflector provides a thermally conductive layer that might be used to induce the PCM phase switch through the use of embedded microheaters [52]. The metallic back plane also helps to reduce the PCM volume required to shift the resonance, in turn improving the melt–quench cooling rates, very high levels of which (up to 10 s of K/ns) are inherently necessary for a successful re-amorphisation process [33,42].

### 2.2. Meta-Atom Design and Analysis Process

Optical performances of the meta-atoms were designed and simulated using the RF module from the finite element analysis software COMSOL Multiphysics (COMSOL Inc., Stockholm, Sweden). Our models employed Floquet boundary conditions on the sides of the meta-atom unit cell to mimic an infinite array of resonators, whereas top and bottom boundaries were truncated with perfectly matched layers to avoid computational reflections. We used a swept triangular mesh, with element sizes chosen by the software according to the sizes of the components. We conducted stationary studies solving the wave equation in the frequency domain. The structures were excited at normal incidence by means of periodic ports, where the complex reflection coefficients (optical amplitude and phase) were also monitored.

Since the resonant frequency of an optical resonator (and therefore the phase and amplitude imposed on an incident wave of light re-radiated to the free-space) is entirely determined by its geometry, as well as by the optical properties of the constituent materials [53], optimisation was carried out via monitoring changes in amplitude and phase upon varying the meta-atom geometrical parameters. Final geometrical parameters were determined to satisfy two conditions simultaneously:A full 2π phase coverage as a function of the resonator radius for one PCM state (in resonance);A nearly flat phase profile as a function of the resonator radius for the other PCM state (out of resonance).

We then used a MATLAB script to build our focusing meta-reflector arrays. The script allocates the radii of the resonators as a function of their local phase in order to satisfy Equation (Equation 1) for a variety of focal lengths. After building our meta-reflector and representing it as a matrix of complex field amplitudes (assuming uniform illumination), the MATLAB script calculates the output intensity. For lower numerical apertures (paraxial regime), we employed the Fresnel approximation, in which the amplitude U(x,y) in the observation plane is given by:(2)$$U\left(x,y\right)=\frac{e^{ikz}}{i\lambda z}e^{i\frac{k}{2z}\left(x^{2}+y^{2}\right)}{\;\int \int \;}_{-\infty }^{\infty }U\left(\xi ,\eta \right)e^{i\frac{k}{2z}\left(\xi^{2}+\eta^{2}\right)}e^{-i\frac{2\pi }{\lambda z}\left(x\xi +y\eta \right)}\;d\xi \;d\eta ,$$
where *U* is the amplitude, ξ and η are the coordinates of the meta-reflector plane, *x* and *y* are the coordinates of the observation plane and *z* is the separation of these two planes [54]. We utilised the 2D FFT function to perform this integral. For larger numerical apertures, the Fresnel approximation is less accurate, and we instead implemented the full Rayleigh–Sommerfeld integral. Following the discussion in chapter 3 of Ref. [54], diffraction may be regarded as a linear system, meaning that the Fourier transform $G_{2}$ of the output (observation plane) amplitude is related to the Fourier transform $G_{1}$ of the input (meta-reflector plane) amplitude by a transfer function *H*:(3)$$G_{2}\left(f_{X},f_{Y}\right)=H\left(f_{X},f_{Y}\right)G_{1}\left(f_{X},f_{Y}\right),$$
where $G_{1}$, $G_{2}$ and *H* are all expressed as functions of the spatial frequencies. For the Rayleigh–Sommerfeld integral, the transfer function is given by:(4)$$H\left(f_{X},f_{Y}\right)=\left\{\begin{array}{cc} \exp(i\frac{2\pi z}{\lambda }\sqrt{1-{\left(f_{X}\lambda \right)}^{2}-{\left(f_{Y}\lambda \right)}^{2}}), & \mathrm{if}\;\sqrt{f_{X}^{2}+f_{Y}^{2}}<\frac{1}{\lambda }\\ 0, & \mathrm{otherwise}. \end{array}\right.$$
We thus propagated our fields by Fourier transforming our input, multiplying the result by this transfer function and inverse Fourier transforming the product.

### 2.3. Thermal Simulations

Thermal simulations were carried out in COMSOL via coupling the RF module to the heat transfer in solids module. Since our devices would usually operate in the continuum (i.e., under constant illumination), a frequency-domain stationary study was carried out.

Our data for the thermal properties of Ge${}_{2}$Sb${}_{2}$Se${}_{4}$Te (the PCM in our study) were taken from Ref. [55]. Properties of Si${}_{3}$N${}_{4}$ were taken from Refs. [56,57,58] and of tungsten from Ref. [59].

## 3. Results and Discussion

### 3.1. NA Control in the Mid-Infrared

Figure 2a shows a schematic of the unit cell employed in our mid-IR devices (design wavelength λ=3000 nm). It consists of a PCM cylindrical resonator lying on a metallic plane. Here, the PCM Ge${}_{2}$Sb${}_{2}$Se${}_{4}$Te (GSST) was chosen due to its high transparency, and high optical contrast between amorphous and crystalline states in the mid-IR [60]. Optical properties of the materials employed in our simulations are shown in Appendix A. The back reflector is made of tungsten, which has good plasmonic properties in this spectral regime, and whose melting point is significantly above the melting point of GSST (so the W layer could potentially be used, in real-world devices, to provide a resistive heating element to induce crystallisation and re-amorphisation of the GSST layer). A thin Si${}_{3}$N${}_{4}$ layer (8 nm) is inserted between the GSST and the back-plane to prevent any diffusion of tungsten into the resonator [42,61].

As described in Section 2.2, the geometrical parameters of the cylinder were optimised to span a 2π optical phase coverage as a function of its radius in one structural state (on-state), and simultaneously delivering a nearly flat optical phase profile in the other state (off-state). In this design, we chose the GSST crystalline state as the on-state due to its high refractive index (see Appendix A), which allows the reduction in the volume of GSST required to excite resonances at the design wavelength. A smaller volume of the PCM results in faster cooling rates, facilitating re-amorphisation during the melt–quench process [62]. As can be seen in Figure 2b,c, with suitable design when the GSST meta-atoms are in the crystalline state, a full 2π (360${}^{\circ }$) phase coverage can be achieved when varying the disk diameter D (Figure 2b) around the resonance (Figure 2c). When switching the GSST to the amorphous state (lower refractive index, see Appendix A), the cylinder becomes too small optically to support a resonance at the excitation wavelength (λ = 3000 nm), exhibiting an out-of-resonance behaviour with a nearly invariant (flat) optical phase profile as a function of the cylinder diameter Figure 2b.

After designing the basic building blocks of our mid-IR meta-reflectors, we simulated devices with two different focal lengths (f = 10 cm and f = 1 cm) with a semi-aperture of radius R = 0.5 cm. The numerical aperture of a focusing reflector is given by the relation of these two parameters:(5)NA=sin(arctan(R/f)).
Therefore, when our devices are in the crystalline state, they have NAs of 0.0499 (f = 10 cm, R = 0.5 cm) and 0.4472 (f = 1 cm, R = 0.5 cm), respectively. Now, by switching the outer ring-like region of the lens to the amorphous state (for example by supplying heat with electrical micro-heaters integrated with the tungsten substrate, similar to the approach in reference [52]), we can effectively reduce the semi-aperture by creating an outer flat phase profile, thus reducing the NA accordingly. In Figure 3a,b, we show the calculated spot sizes for both focal lengths with gradually decreasing NAs. In line with diffraction theory, as can be seen from our results, decreasing the semi-aperture results in a decrease in NA and subsequent spreading of the focal spots for both focal lengths, as quantitatively shown in Table 1 (f = 10 cm) and Table 2 (f = 1 cm). We observe an unaltered spot shape for NAs down to 0.0290 in the first case and to 0.2607 in the second. Intensity profiles of these focal spots are displayed in Figure 3a,b. We follow reference [63] and define the focusing efficiency as the ratio of the power within a spot of diameter equal to three times the simulated full width at half-maximum (FWHM) to the total incident power. Our efficiencies at around 14% maximum are comparable, but somewhat lower, than other phase-change meta-lenses in the literature [12,35,36,37,38], but as we pointed out in the introduction, other reported approaches use polarisation-sensitive geometric phase and/or rather complex or high aspect ratio meta-atoms that could make re-amorphisation of the PCM volumes difficult.

Apart from increasing the spot size, decreasing the NA should also result in an increase in the depth of focus. To investigate this, we also simulated the intensity along the optical axis. In Figure 3c,d, we show results from such simulations for a focal length of f = 10 cm, and two different NAs of 0.0499 (Figure 3c) and 0.0300 (Figure 3d). From these plots, we can clearly see how, by reducing the NA, the focal position remains stationary, whereas both the spot size and the depth of focus increase, as expected.

From Figure 3c,d it can be seen that there is also an apparent ‘secondary focus’ at half the intended focal length. We attribute this to the following effect: our meta-atoms are arranged to yield the parabolic phase profile described by Equation (1) and schematically illustrated in Figure 1b. Moving out from the centre of the device, the spacing of successive maxima in this pattern decreases, as does the rate of this change (as dφ/dr and $d^{2}\varphi /dr^{2}$, where the radius $r=\sqrt{x^{2}+y^{2}}$, are both negative). This leads to the pattern becoming approximately periodic in the outer part of the device, with an average period longer than the wavelength. This, in turn, produces diffractive effects in the outer part of the device, leading to some light being lost into higher diffraction orders. We model this effect as a diffraction grating occupying the region *r* = 0.3–0.5 cm, whose grating constant is the average of the separation of the phase maxima in this region, which we find to be 74 μm. This gives a first diffraction order at 0.04 rad to the normal, which corresponds to the true focus, and a predicted second order at 0.0812 rad. Where the second order beams from the whole circumference of the device intersect on the optical axis, an apparent second focus would emerge. The actual ‘secondary focus’ in our simulation is at an angle of 0.0798 rad, extremely close to that expected according to the above description. Further diffraction orders are also present, but their intensity is negligible. This effect reduces the efficiency of the devices, but it does not hinder their main function.

### 3.2. Thermal Performance of Mid-IR Meta-Reflectors

For high-power applications, it is possible that the constituent materials of devices may be heated to their melting point (or, in the case of an amorphous GSST, to its crystallisation point), placing a limit on the maximum intensity for which a device can be used for its intended application. We investigated this threshold in our mid-IR designs by simulating the conversion of absorbed electromagnetic energy to heat and the spread of heat in the device in the continuum (constant illumination).

For this purpose, we first found the maximum temperature reached in the PCM layer for different resonator radii and for a fixed irradiance intensity of I = 10${}^{9}$ Wm${}^{-2}$ (Figure 4a). For the crystalline state (red lines in Figure 4), the diameter producing the highest temperature within the GSST cylinder is indeed the one coinciding with the in-resonance condition (D = 710 nm), which corresponds to a drop in reflectivity (as shown in Figure 2c from the previous section), and therefore to more light being absorbed by the surface and converted into heat. Since amorphous GSST has essentially no optical losses at 3 μm (see Appendix A), no significant temperature increase, under constant illumination, takes place for any of the cylinder diameters employed in our designs, when the PCM layer is in its amorphous state. The maximum temperature for the amorphous state is reached for a cylinder diameter of D = 1300 nm (blue line) and amounts to only 296.4 K.

Having identified the diameters producing a maximum temperature increase, we then fix the diameter and make a sweep over the incident intensity to find the crystallization and melting thresholds. By comparing the maximum temperature within the different material layers of our model (namely, GSST, W and Si${}_{3}$N${}_{4}$ shown in Figure 4b–d, respectively) with the melting point of the relevant material (903 K for GSST [64], 3695 K for W [65] and 2170 K for Si${}_{3}$N${}_{4}$ [66]), we find that our device should be able to withstand laser intensities up to the order of 1.2 × 10${}^{9}$ Wm${}^{-2}$ until crystalline GSST resonators reach the melting point (Figure 4b, blue line). The temperature increase in the amorphous phase at the same intensity is negligible (Figure 4b, red line); thus, no crystallisation of amorphous meta-atoms is expected below the melting threshold imposed by crystalline meta-atoms. All other materials in our design (W and Si${}_{3}$N${}_{4}$) stay far below their respective melting points for all intensities used here, see Figure 4c,d.

Among common uses of CW lasers at these wavelengths, medical applications might employ powers of the order of 1–10 W [67,68], while trace gas detection can operate with 5–100 mW [69,70]. Therefore, the former powers could be managed by our devices if the beam cross-section is at least of the order of 1 mm${}^{2}$, and the latter could work down to an area of 10 μm${}^{2}$. Cutting of non-metals is performed with mid-IR lasers with kilowatt powers or more [71]. Application in this domain would be harder but could still work if the beam covers the entire 1 cm diameter of our simulated devices.

### 3.3. Metasurfaces for the Visible

Since the potential applications of our concept are not limited to the infrared, we also propose and simulate a design for operation at a wavelength of 632.8 nm. As shown in Figure 5a, for this design the phase-change alloy chosen is Sb${}_{2}$S${}_{3}$, it being the PCM with one of the highest bandgaps reported so far (thus better transparency in the visible spectrum) [72]. In this case, the PCM layer is sandwiched between two TiO${}_{2}$ layers to minimise the volumes of Sb${}_{2}$S${}_{3}$ required to excite a resonance at the design wavelength [41] (note that this was not necessary for the mid-IR design, because GSST re-amorphises more easily than Sb${}_{2}$S${}_{3}$ [60,73]).

As shown in Figure 5b, and contrary to the mid-IR design (where the on-state is the crystalline phase), for our visible range design we chose the amorphous state as the on-state to provide a full 2π optical phase control as a function of the cylinder diameter. This is justified by the fact that the optical contrast of Sb${}_{2}$S${}_{3}$ at a wavelength of 632.8 nm is significantly worse than that of GSST at 3000 nm (see Appendix A). Therefore, the resulting shift in resonance is not enough to fully suppress optical phase variations as a function of the cylinder diameter; thus, no design directly analogous to the mid-IR design can be achieved in the visible. Instead, we take advantage of the high optical losses of the Sb${}_{2}$S${}_{3}$ crystalline state (see Appendix A) to change the resonance to the overdamped regime, where only small variations of the optical phase can occur around the resonance [74], as revealed by Figure 5b,c (blue lines).

In line with our previous design in the mid-IR, we simulated meta-reflectors in the visible having different numerical apertures (see Table 3), and calculated their far-field intensity patterns accordingly. The intensity profiles are shown in Figure 5d, revealing a gradual increase in the spot size when reducing the NA (i.e., closing the aperture). Our design platform is therefore completely scalable to smaller wavelengths via proper material selection and geometry re-optimization.

## 4. Conclusions

We have successfully designed and simulated the performance of chalcogenide-based phase-change meta-lenses (strictly, focusing meta-mirrors) for dynamic control of numerical aperture. Due to the capability of chalcogenide PCMs for ultra-fast switching between states, our meta-lenses offer a potential route to the ultra-fast, non-volatile control of numerical aperture, all in a compact, lightweight and flat format. Our design is especially suited to mid-IR wavelengths, where the optical absorption of PCMs can be low. Indeed, at a wavelength of 3 μm, we were able to adjust the NA of both large- and small-aperture Ge${}_{2}$Sb${}_{2}$Se${}_{4}$Te-based devices by factors of 1.72 and 1.85, respectively. Sb${}_{2}$S${}_{3}$-based designs for use in the visible part of the spectrum were also successfully developed, where NA control factors of 1.92 were achieved (though with a reduced accessible NA range). Our design approach is, in principle, scalable to other laser wavelengths; the only requirement is that a PCM exists with sufficiently good optical properties to achieve the efficiencies needed for a particular application.

## Figures and Tables

**Figure 1 nanomaterials-12-02689-f001:**
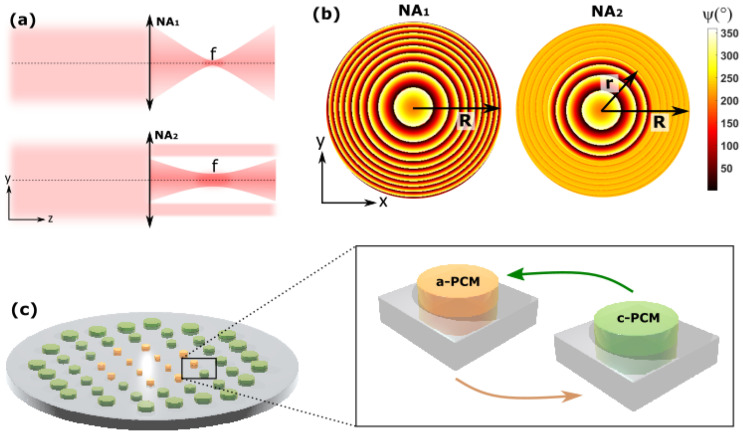
(**a**) Effect of NA switch on an incident beam, with the larger NA value above and the smaller below. The reduction in NA is accompanied by a broadening of the focal spot. Note that our device operates in reflection, but this illustration is shown in transmission for clarity. (**b**) Illustrative phase profile of device in larger NA state on left (whole device of radius R with parabolic phase profile) and smaller on right (inner region of radius r with parabolic phase profile and outer region with near-flat phase profile). (**c**) Schematic illustration of device with the inner region shown in yellow and the outer in green. The switch is effected by changing the material phase of the outer region phase-change materials (PCMs).

**Figure 2 nanomaterials-12-02689-f002:**
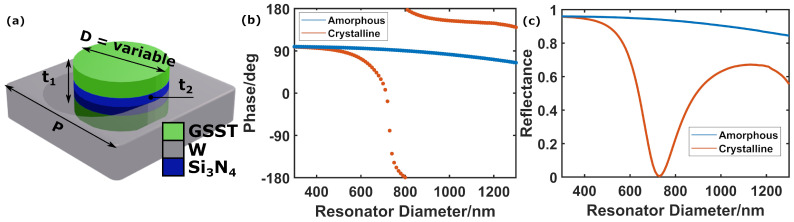
(**a**) Schematics of the mid-IR unit cell or meta-atom. Optimum geometrical parameters after optimisation were found to be *p* = 1400 nm, t${}_{1}$ = 179 nm and t${}_{2}$ = 8 nm. (**b**) Dependence on resonator diameter of the phase imparted by the meta-atom on reflected light. (**c**) Dependence on resonator diameter of the meta-atom reflectance.

**Figure 3 nanomaterials-12-02689-f003:**
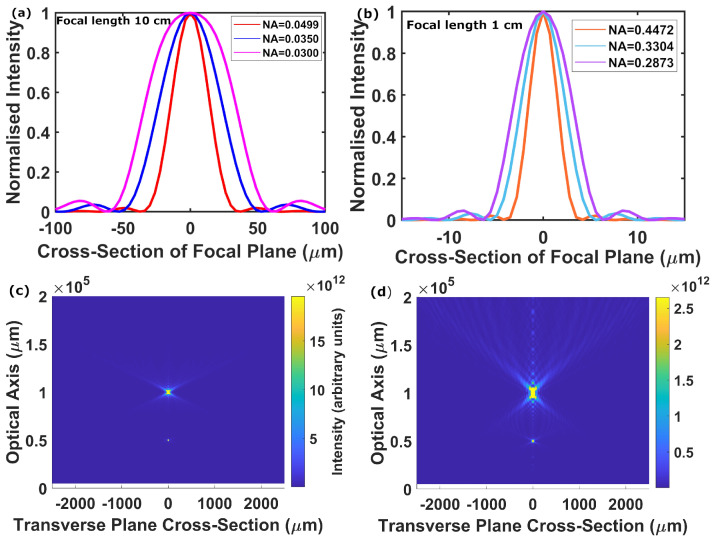
(**a**,**b**) Intensity profiles of focal spots generated by the 3000 nm wavelength design. The profiles shown are for the entire device in the crystalline phase (full semi-aperture), and for two gradually decreasing semi-apertures achieved via switching the outer ring-like region to the amorphous state. The profiles are all normalised to the same scale to allow easier comparison of the shapes. (**a**) Red is NA = 0.0499 (R = 0.5 cm), blue is NA = 0.0350 (R = 0.35 cm), magenta is NA = 0.0300 (R = 0.30 cm). (**b**) Orange is NA = 0.4472 (R = 0.5 cm), light blue is NA = 0.3304 (R = 0.35 cm), purple is NA = 0.2873 (R = 0.30 cm). (**c**,**d**) Simulated intensity profiles along the optical axis and in a cross-section of the transverse plane for a focal length of 10 cm, revealing clear differences in both the spot size and the depth of focus. (**c**) Full device crystalline, NA = 0.0499. (**d**) Outer region amorphous, NA = 0.0300.

**Figure 4 nanomaterials-12-02689-f004:**
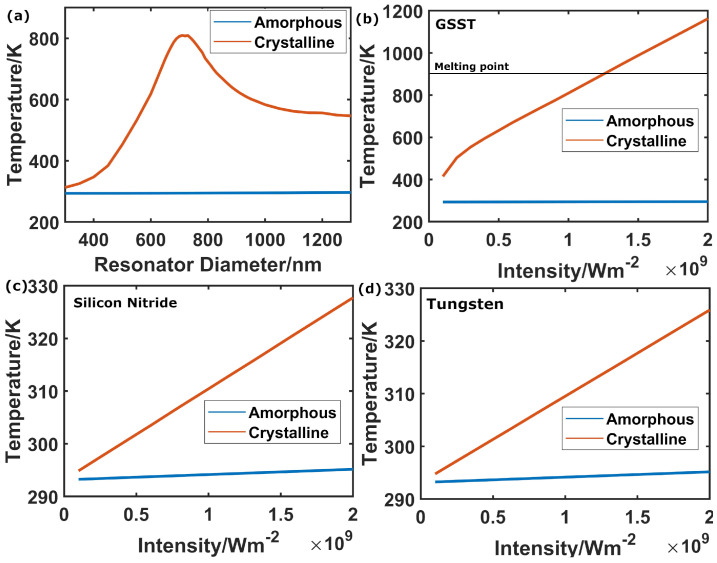
(**a**) Maximum temperatures reached in our model as a function of resonator radius at an incident intensity of 10${}^{9}$ Wm${}^{-2}$ with the GSST in both amorphous (blue) and crystalline (red) phases. The maxima occur at 1300 nm diameter for the amorphous and 710 nm for the amorphous. (**b**–**d**) Maximum temperatures in our model as a function of incident intensity in the (**b**) GSST, (**c**) Si${}_{3}$N${}_{4}$ and (**d**) W layers with the GSST in both amorphous (blue) and crystalline (red) phases. The GSST melting temperature of 903 K imposes the threshold, marked with the horizontal line in (**b**). The melting temperatures of the other materials are beyond the ranges of the respective graphs.

**Figure 5 nanomaterials-12-02689-f005:**
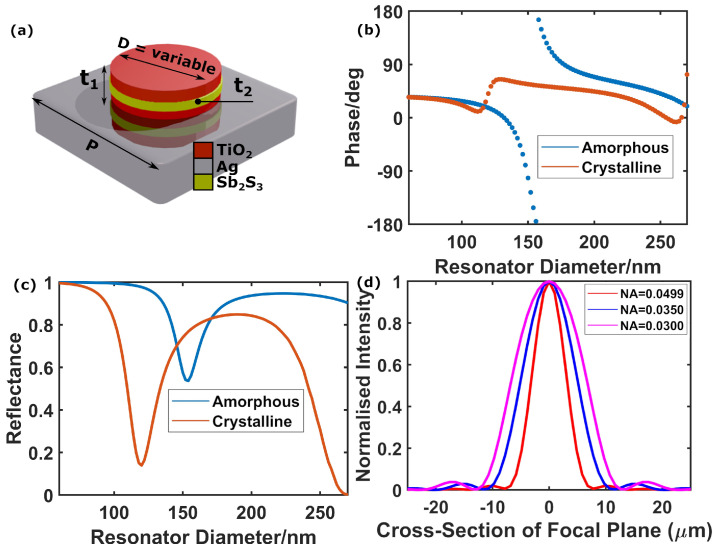
(**a**) Meta-atom design for 632.8 nm wavelength operation, with a Sb${}_{2}$S${}_{3}$ layer centred between two TiO${}_{2}$ layers on a silver back-plate. Geometrical parameters are here *p* = 370 nm, t${}_{1}$ = 74 nm and t${}_{2}$ = 32 nm. (**b**,**c**) Optical phase (**b**) and amplitude (**c**) variation as a function of the resonator diameter D for amorphous (blue) and crystalline (red) states. (**d**) Intensity profiles of focal spots generated by the 632.8 nm wavelength design with focal length 10 cm. The profiles shown are for the entire device in the crystalline phase (NA = 0.0499, red), and for amorphous inner regions of radii of 0.35 cm (NA = 0.0350, blue) and 0.30 cm (NA = 0.0300, magenta). The profiles are all normalised to the same scale to allow easier comparison of the shapes.

**Table 1 nanomaterials-12-02689-t001:** Simulated data for the 3000 nm devices with focal length 10 cm.

NA	Semi-Aperture/cm	FWHM/μm	Peak Intensity (au)	Focusing Efficiency
0.0499	0.50	30.9	258	13.9%
0.0350	0.35	49.9	48.4	6.82%
0.0300	0.30	70.0	19.9	5.25%
0.0290	0.29	75.5	16.3	4.91%

**Table 2 nanomaterials-12-02689-t002:** Simulated data for the 3000 nm devices with focal length 1 cm.

NA	Semi-Aperture/cm	FWHM/μm	Peak Intensity (au)	Focusing Efficiency
0.4472	0.50	3.41	88.8	13.7%
0.3304	0.35	5.11	19.1	6.68%
0.2873	0.30	6.71	8.94	5.26%
0.2670	0.27	8.36	5.13	4.37%

**Table 3 nanomaterials-12-02689-t003:** Simulated data for the 632.8 nm device.

NA	Semi-Aperture/cm	FWHM/μm	Peak Intensity (au)	Focusing Efficiency
0.0499	0.50	6.56	22.98	13.7%
0.0350	0.35	10.4	4.68	6.82%
0.0300	0.30	13.7	2.21	5.33%
0.0260	0.26	18.0	1.07	4.09%

## Data Availability

Data supporting results discussed in this manuscript are openly available in the University of Exeter’s repository, ORE.

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
