# Peer review of "Optical and Thermal Design and Analysis of Phase-Change Metalenses for Active Numerical Aperture Control"

_nanomaterials, 2022, doi:10.3390/nano12152689_

Round 1
Reviewer 1 Report
In this paper, George Braid et al. present the design of meta-lenses with dynamic NA control based on phase-change materials, exploring two different spectral regimes. A change of NA by a factor of 1.92 was demonstrated. These results are interesting for the study of active photonics. However, the authors need to address the following questions and concerns before the publication in Nanomaterials.
1. High NA is a long-term goal for designing optical lenses or metalenses. However, NA is very low in this work (0.026-0.049 in visible range). The authors need to explain it and discuss how to increase the NA.
2. Phase-change metasurfaces or nanoantennas are hot topic recently. In Introduction, the authors have presented enough literatures on GST. While for the other important phase change material, VO2, there is no discussion and reference. Some papers need to be cited (e.g., Nanoscale Horizons, 2019, 4, 712-719; ACS Photonics 2021, 8, 1048–1057), and additional discussion is necessary to compare GST and VO2.
3. Why the dielectric resonators are designed on metallic substrates? Lenses are supposed to be transparent. As discussed in Page 5, has the plasmonic property of W played an important role on optical phase engineering?
4. In Page 5, the statement: “by switching the outer ring-like region of the lens to the amorphous state” needs additional discussion on how to realize in experiments.
5. As for the thermal performance shown in Fig. 4, tungsten shows the smallest increase of temperature which is unreasonable. The ultrathin gap (tSi3N4=8nm) between GSST and W can support localized surface plasmon resonances and generate considerable local heating.
6. In “3.3 Metasurfaces for the visible”, the authors change the working wavelength, the PCM, the structures…, which lose the logical coherence compared with above part.
Reviewer 2 Report
In my opinion, this work is of obvious importance to the designing and creating of future functional meta-lenses (focusing meta-mirrors) with dynamic NA control based on phase-change material. This paper presents the design of focusing meta-mirrors for two frequency ranges - visible (based on Sb2S3) and mid-infrared (based on Ge2Sb2Se4Te). This work could be improved only by experimental confirmation of the calculations performed. In my opinion, the paper is quite detailed and provides all the comprehensive data. I do not consider it necessary to make any critical remarks. There is no doubt that this paper is very useful for the nanophotonics and optoelectronics communities. I think that the novelty of this paper is sufficient to be published in Nanomaterials.
A few remarks:
The paper contains a table with the optical constants of the materials used, but there is no reference to the articles from which they were taken.
(anywhere in the text) Ge2Sb2Se4Te1 should be replaced by Ge2Sb2Se4Te.
(line 124) GeSbSeTe should be replaced by Ge2Sb2Se4Te.
Reviewer 3 Report
The paper’s idea is interesting. However, details regarding possible technology processes and acceptable fabrications errors are required to increase the utility.
The spectral characteristics of the meta-atoms strongly depend on the shape and size of the nanostructure.
You investigated the influence of the diameter on the resonance, but also the influence of the shape has to be analyzed.
It is impossible to obtain perfect cylinders, with the designed parameters, even in case of a single layer- and you have multi-layered structures. The final shape can be a kind of truncated cone or, in case of multi-layer structure the shape can be more complex as each layer has a different etching rate. The small changes of the shape will lead to significant shift in resonance, or multiple resonance, especially in visible range.
Reviewer 4 Report
This paper cannot be accepted for publication unless the authors can show that this idea is something novel or they showed something which the authors in pt. 5 below do not show.
At this stage, I prefer to reject the paper since the number of modifications that need to be made will substantially result in a very different paper altogether.
1. The title of the manuscript needs to be changed to highlight that it is a "simulation" paper otherwise the paper is misleading.
2. The authors need to provide a table benchmarking their results with the state-of-art results.
3. How is the focussing efficiency calculated and what are its values. Please refer to a standard metric defined in this paper to calculate the same: https://opg.optica.org/optica/fulltext.cfm?uri=optica-6-6-805&id=413582
4. When the authors write about imaging applications, they should show imaging results (even simulated imaging results if possible). Otherwise, simply reporting FWHM values does not prove anything; rather the authors should report the "Strehl ratio" values for these metalenses.
5. The authors leave out a lot of relevant examples of tunable metalenses. I cannot say that their paper is novel.
Here are a few:
(a) https://www.nature.com/articles/s41598-019-41859-x#:~:text=The%20metalens%20is%20composed%20of,is%20as%20high%20as%200.71.
(b) https://opg.optica.org/oe/fulltext.cfm?uri=oe-29-5-7925&id=448636
(c) https://dspace.mit.edu/handle/1721.1/142626
(d) https://www.mdpi.com/2076-3417/9/22/4927
(e) https://www.frontiersin.org/articles/10.3389/fphy.2021.651898/full
Round 2
Reviewer 1 Report
I would like to thank the authors for addressing all my comments.
Reviewer 3 Report
Although the author's reply is not fully convincing (the cylinders already fabricated larger than those proposed in this paper for visible range and are far from perfect) , I think the revised version with the new title is acceptable. The authors have the possibility to fabricate the device and experimentally verify the proposed design.
Reviewer 4 Report
The authors made satisfactory revisions to the manuscript. I can recommend the publication of the manuscript in current form.